# Transcriptome Profiling of *Rhipicephalus annulatus* Reveals Differential Gene Expression of Metabolic Detoxifying Enzymes in Response to Acaricide Treatment

**DOI:** 10.3390/biomedicines11051369

**Published:** 2023-05-06

**Authors:** Amritha Achuthkumar, Shamjana Uchamballi, Kumar Arvind, Deepa Azhchath Vasu, Sincy Varghese, Reghu Ravindran, Tony Grace

**Affiliations:** 1Department of Genomic Science, School of Biological Sciences, Central University of Kerala, Kasaragod 671320, Kerala, India; amrithaasha@gmail.com (A.A.);; 2Department of Biochemistry, Pazhassiraja College, Pulpally 673579, Kerala, India; 3Department of Veterinary Parasitology, College of Veterinary and Animal Sciences, Pookode 673576, Kerala, India

**Keywords:** ticks, *Rhipicephalus* (*Boophilus*) *annulatus*, acaricides, amitraz, resistance, bioassays, transcriptome sequencing, cytochrome P450

## Abstract

Ticks are hematophagous ectoparasites of economic consequence by virtue of being carriers of infectious diseases that affect livestock and other sectors of the agricultural industry. A widely prevalent tick species, *Rhipicephalus* (*Boophilus*) *annulatus*, has been recognized as a prime vector of tick-borne diseases in South Indian regions. Over time, the use of chemical acaricides for tick control has promoted the evolution of resistance to these widely used compounds through metabolic detoxification. Identifying the genes related to this detoxification is extremely important, as it could help detect valid insecticide targets and develop novel strategies for effective insect control. We performed an RNA-sequencing analysis of acaricide-treated and untreated *R.* (*B.*) *annulatus* and mapped the detoxification genes expressed due to acaricide exposure. Our results provided high-quality RNA-sequenced data of untreated and amitraz-treated *R.* (*B.*) *annulatus*, and then the data were assembled into contigs and clustered into 50,591 and 71,711 uni-gene sequences, respectively. The expression levels of the detoxification genes across different developmental stages of *R.* (*B.*) *annulatu* identified 16,635 transcripts as upregulated and 15,539 transcripts as downregulated. The annotations of the differentially expressed genes (DEGs) revealed the significant expression of 70 detoxification genes in response to the amitraz treatment. The qRT-PCR revealed significant differences in the gene expression levels across different life stages of *R.* (*B.*) *annulatus*.

## 1. Introduction

Ticks are vectors that have a worldwide distribution and are of huge economic relevance due to the direct harm they inflict on their hosts. Ticks acquire pathogens from infected vertebrate hosts and transmit diseases to other animals through blood meal [1]. As a result, ticks have an impact on public health due to several human diseases that they actively transmit, such as Lyme disease, Kyasanur forest disease, tick-borne encephalitis, rickettsial diseases, and several other zoonotic illnesses [2,3].

Due to the increasing cases of cattle tick infestations in tropical and subtropical regions, tick control is a priority for developing countries [4]. Tick infestation depletes milk production and as such, they are among the most harmful vectors of lethal pathogens, second only to mosquitos [5]. Farm animals can be infected by a number of tick species that multiply inside the host and cause damage through direct injury or via the transmission of diseases.

*Rhipicephalus* (*Boophilus*) *annulatus* (*R.* (*B.*) *annulatus*) is a widespread exophilic tick species that live and feed on the host for several days; they ingest blood meal by attaching their hypostome to the skin of the cattle. Once fed, ticks commence engorgement, molting, reproduction, and finally, they die [6]. *R.* (*B.*) *annulatus* infestation has a negative impact on milk productivity due to the induced stress, immune dysfunction, blood loss, and the transmission of pathogens such as *Babesia bovis* and *Babesia bigemina* [7,8]. *R.* (*B.*) *annulatus* is a member of the family *Ixodidae* and pose a threat to livestock in South India due to their capacity to spread many variants of tick-borne diseases, including anaplasmosis and babesiosis in cattle [9,10]. The incidence and prevalence of the ixodid vector species were reported in domestic animals in the South Indian states of Kerala [11,12] and Tamil Nadu [13]. Exclusively, obligate hematophagous ectoparasitic ticks infect approximately 80% of the world’s cattle population and are among the leading cause of production loss in the livestock sector and dairy industry [14]. The outbreaks of tick-borne infectious diseases in the Indian states of Kerala, Gujarat, and Punjab have emphasized the importance of controlling ticks [15,16,17,18].

In the field, commercially available vaccines have been effective at enhancing the immunity of cattle against ticks [19,20]. However, vaccine efficacy varies due to strain variation; for instance, a vaccine developed for Australian tick strains might not be effective for targeting Brazilian tick strains. Similarly, the recombinant Bm86 vaccine has induced different immunological responses in *R.* (*B.*) *microplus* and *R.* (*B.*) *annulatus* [21,22] that indicated that genetic and/or physiological differences could be the cause of the variation in vaccine efficacy. Despite the many years of dedicated searching for tick-specific antigens for tick vaccines, only a limited number of vaccines have been developed against *Ixodidae* [23,24,25,26]. This functional variability and limited vaccine availability has limited the ability to provide full protection for cattle and eradicate cattle infestation.

In this context, chemical acaricides continue to be a control method for cattle producers to mitigate tick infestations. The availability and ease of handling has made acaricide treatment the dominant method for tick control. However due to their frequent use these acaricide residues have been shown to contaminate the environment [27]. The long-term use of chemicals promoted the development of acaricidal resistance that increased the survival ability of ticks [28,29,30,31,32].

Being one host ticks, *R.* (*B.*) *annulatus* can easily develop resistance properties to many classes of acaricides. *R.* (*B.*) *annulatus* have been constantly challenged by the repeated usage of chemical acaricides, and as a result, a vast array of defenses has been employed, including biochemical defenses such as AchE insensitivity, sodium channel insensitivity, and GABA receptor (Cl—channel) insensitivity.

In many cases, ticks express a different set of genes at high levels to counter acaricide treatments that could be used as a target to further develop efficient tick control approaches [33,34,35]. The detoxification of acaricides via the enhanced activity of enzyme families of Cytochrome P450 (CYP450), Esterases (ESTs), Glutathione-S-transferase (GST), and ATP Binding Cassette (ABC) transporters has been well reported in *Ixodes scapularis* [36], *Boophilus microplus* [37], *Dermacentor variabilis* [38], and *Rhipicephalus sanguineus* [39] through RNA sequencing. However, extensive research has yet to evaluate the metabolic enzyme-mediated detoxification mechanisms and the corresponding resistance in *R.* (*B.*) *annulatus*. Exploring the interaction between the detoxification genes and insecticide resistance could help with the development of integrated pest management programs, thereby defining the precise role of these genes in the development of acaricide resistance in *R.* (*B.*) *annulatus*.

A formamidine acaricide, amitraz, was initially introduced alongside synthetic pyrethroids in 1986 to restrain organophosphate-resistant ticks, but its usage was limited due to the high cost. The resistance mechanism of *R.* (*B.*) *microplus* [33,40,41,42] and *R. decoloratus* against amitraz [43] has been experimentally investigated around the world. These studies demonstrated a substantial increase in the activity of metabolic detoxification enzymes, such as cytochrome P450s and glutathione-S-transferase, against amitraz. There have been ample reports highlighting the amitraz-resistance mechanism, and molecular studies have highlighted the cytotoxic effects of amitraz [44] and the in vitro efficacy of the compound in various tick populations [45]. However, this has not yet been replicated in *R.* (*B.*) *annulatus* populations.

In this study, we leveraged a transcriptomic approach to identify and quantify the transcripts of *R.* (*B.*) *annulatus* larvae following amitraz treatments and compared the results with a non-treated control group. The most likely candidates for the differentially expressed detoxification genes were considered for validation using qRT-PCR. We demonstrated the differences in the expressed genes/transcripts across different developmental stages of *R.* (*B.*) *annulatus.* In addition, a comparison of the gene expression profiles associated with treatments of amitraz and flumethrin (a commonly used pyrethroid) was performed. The information revealed through this study could enhance our understanding and help in the development of potential strategies for controlling this menacing pest.

## 2. Materials and Methods

### 2.1. Collection and Rearing of Ticks

The adult engorged female ticks were obtained from the Instructional Farm, Kerala Veterinary and Animal Sciences University, Wayanad, Kerala. The ticks were collected from infested cattle and maintained in a biochemical oxygen demand (BOD) cabinet with 80 ± 5% relative humidity. Under this incubation condition, ticks were allowed to oviposit, with subsequent hatching into larvae. It took roughly 20 days for egg laying, 10–15 days for the eggs to hatch into larvae, and another 20 days for the larvae to mature into an adult. Both larvae (8–10 days old) and newly engorged adult ticks were used for further experiments.

### 2.2. Larval Packet Test

The larval packet test (LPT) was performed based on the Food and Agricultural Organization (FAO’s) standardized protocol (1971) [46] with slight modifications. For the study conducted to determine the toxicity of different solvents for the acaricides, methanol was the least toxic [47]. Thus, for dose-dependent bioassays, stock solution was prepared by dissolving the active ingredient of amitraz and flumethrin in methanol. This formulation was sequentially diluted in methanol to generate series concentrations that ranged from 0.146 ppm to 300 ppm for amitraz and 0.024 ppm to 25 ppm for flumethrin. Three replicates were used to estimate the relevant concentration of chemical acaricides by LPT. Larvae were exposed to the acaricide by allowing the solution containing required concentration of acaricide to be impregnated on filter paper folded into a triangular packet shape and then allowing them to dry. Approximately 100 live larvae aged 10–12 days were kept in each impregnated packet and placed in the BOD (Bio-Oxygen Demand) incubator at 28 ± 10 °C, 85 ± 5% RH for 24 h. The packets were opened following the incubation period and the mortality percentage was calculated. The LC_50_, which is the median lethal concentrations of both acaricides, was assessed in *R.* (*B.*) *annulatus* using the Abbott formula.

### 2.3. Adult Immersion Test

The stage specific expression analysis was performed by comparing the acaricide treated larval and adult tick populations. The optimum acaricide concentration for the treatment of adult ticks was obtained through the Adult Immersion Test (AIT). The AIT was based on the procedure detailed in [48]. Fully engorged adult female *R.* (*B.*) *annulatus* were doused in distilled water and drained on absorbent paper before being used for AIT. The different concentrations of amitraz (200 ppm–350 ppm) and flumethrin (20 ppm–100 ppm) were prepared using methanol, as described above. Four replicates, containing six ticks each, were immersed in the respective concentration for two minutes. After two minutes, ticks were transferred to plastic tubes and kept in a BOD incubator with 80% relative humidity at 28 °C. The specimen tubes were observed for oviposition, and mortality was assessed up to 15 days. The egg mass was continuously monitored in the BOD incubator, under the same incubation state, for 30 days.

### 2.4. Total RNA Extraction and RNA Sequencing

Total RNA extraction was performed from the *R.* (*B.*) *annulatus* larvae untreated or treated with amitraz at 0.56 ppm using the trizol RNA isolation protocol (Life Technologies, Carlsbad, CA, USA). The purity and concentration of the total RNA of *R.* (*B.*) *annulatus* was determined using a Nanodrop spectrophotometer 2000 (Thermo Fisher Scientific, Wilmington, DE, USA). To obtain transcriptome data, a cDNA library was constructed using the procedure from the Illumina TruSeq stranded mRNA library prep kit. The cDNA library sequencing was conducted on the Illumina HiSeq 500 platform at the Genotypic technology’s Genomics facility (Genotypic Inc., Bangalore, India). The RNA quantity was measured using a Qubit fluorometer and validated using a high-sensitivity Bioanalyzer chip (Agilent Technologies, Santa Clara, CA, USA). Then, cDNA libraries were prepared for paired-end sequencing on the Illumina HiSeq 500 platform. The raw RNA-seq data for *R.* (*B.*) *annulatus* were deposited into the SRA database (NCBI) and assigned the accession number SRP235650.

### 2.5. Transcriptome Assembly

Quality checks for the raw data (i.e., reads) was achieved using FASTQC software. Low-quality reads were filtered out using NGS QC Toolkit [49] with a stringent Phred-score, ensuring high-quality reads were selected. These high-quality sequences were mapped to form longer contigs using Trinity Assembler [50]. To fill the intra-scaffold gaps, the paired-end reads were used for de novo assembly. The assembled sequences were further processed to obtain unigenes using CD-hit. Similar transcripts and assembled transcripts were annotated against the arthropod’s protein sequences from UniProt. The transcripts with ≥300 bps were considered for further analysis.

### 2.6. Annotation of Sequenced Transcriptome Data

For annotation, BLASTX was performed against arthropod protein sequences from UniProt, followed by NCBI non-redundant databases. Standalone Gene ontology (GO) was performed using the BLAST2GO program [51] and pathways were mapped using the KAAS server (Kyoto Encyclopedia of Genes and Genomes (KEGG) Automatic Annotation Server) [52]. GO terms were assigned to each *R.* (*B.*) *annulatus* transcript and functionally classified into different categories, such as cellular component, biological process, and molecular function. In addition, the *R.* (*B.*) *annulatus* transcriptome data were screened to identify simple sequence repeats (SSRs) using MIcroSAtellite Identification *Tool* (MISA) [53].

### 2.7. Differential Expression Analysis Using RNAseq Method

Expression profiling was performed to offer insight into the genes driving acaricide resistance by classifying the Differentially Expressed Genes between amitraz treated and untreated *R.* (*B.*) *annulatus* larvae. The gene expression level of untreated ticks was used to normalize the gene expression of treated ticks. To gauge the expression level of each unigene (amitraz treated and untreated), the CD-hit program was used to cluster quality reads from each set of samples and to compare gene expression. Differentially expressed genes were identified in treated versus untreated samples; these were further analyzed using DESeq software and Bioconductor (http://bioconductor.org/ (accessed on 25 August 2015) R-packages [54]. Clustered heatmap analysis was performed to visualize gene expression levels of specific contigs by comparing R plots. Finally, the expression levels were quantified between treated and untreated libraries. Additionally, differentially expressed contigs were manually scanned to map the detoxifying genes.

### 2.8. Real-Time PCR

To identify resistant genes in adult ticks and larvae, we used Real-Time PCR analysis. Due to the large size differences between adult ticks and larvae, the drug dosage for the gene expression studies was determined by their respective LC_50_ to normalize the drug exposure to body surface area/volume.

Some of the candidate genes were selected for expression validation using real time PCR. The RT-qPCR was performed with the Roche-Light Cycler 480 System, which is a high-performance, high-throughput PCR platform. The versatile Light cycler 480 instrument provides fast, highly sensitive, and reproducible gene expression results. We selected 7 CYP450 genes and 2 GST genes, CYP (30980), CYP (27576), CYP (51361), CYP (38633), CYP (41222), CYP (48268), CYP (36267), GST (34065), and GST (55171), based on the fold differences pattern from the transcriptome data for the qRT-PCR assay. Expression analysis of the nine identified detoxifying genes in the two developmental stages (larvae and adult) in response to the two separate groups of acaricides at its median lethal concentration (amitraz and flumethrin) was also tested using quantitative real time PCR. The expression of each sample was normalized by the reference gene Beta actin. Three biological replicates were developed for each gene, and all samples were assayed in technical triplicates. The abundance of expressed genes in the acaricide treated and untreated ticks was measured using the 2^−ΔΔCt^ method [55]. The results from the qRT-PCR experiment are presented as a fold change in expression level in drug-treated ticks relative to the gene expression level in untreated ticks. The significant expression of the target gene was statistically calculated using Unpaired *t*-tests.

## 3. Results

### 3.1. Larval Packet Test

The median lethal concentrations (LC_50_) value of amitraz and flumethrin against *R.* (*B.*) *annulatus* was calculated using the larval packet test (LPT). Concentrations ranging from 300 ppm to 0.146 ppm were used to find the lethal concentration of amitraz to *R.* (*B.*) *annulatus* larvae. Bioassays on *R.* (*B.*) *annulatus* with amitraz revealed 100% mortality at 300 ppm and 150 ppm, whereas insignificant mortality was observed in the untreated sample (Appendix A). The mortality observed in the treated group was corrected with the natural mortality occurring in the untreated group using Abbott’s (1952) formula [56]. Corrected mortality data were transformed into their corresponding probits. The dosage–mortality curve LC was plotted after probit analysis [57], and the LC_50_ value was estimated (Figure 1). At 0.56 ppm, 50% mortality was observed in *R.* (*B.*) *annulatus* when treated with amitraz. The ticks that survived the treatment at the LC_50_ concentration were taken for RNA isolation and transcriptome analysis.

For drug treatment with flumethrin, *R.* (*B.*) *annulatus* responded highly even at the lowest concentration. The toxic interaction of flumethrin with *R.* (*B.*) *annulatus* showed 100% mortality when the dose concentration was between 25 ppm and 1.5625 ppm (Appendix A). A graph was plotted using the mortality data against the dosage of flumethrin, and the LC_50_ value was calculated using probit analysis (Figure 2). The concentration of flumethrin needed to kill half of the test population was 0.044 ppm. After determining the median lethal concentration, larvae were exposed to each acaricide at its LC_50_. The surviving larvae at the LC_50_ dose were used for gene expression studies.

### 3.2. Adult Immersion Test and Pattern of Mortality

The two acaricides, amitraz and flumethrin, were also tested against the adult stage of *R.* (*B.*) *annulatus*. The higher concentrations of 200 ppm to 350 ppm were used for amitraz and 20 ppm to 100 ppm for flumethrin. At a concentration used in field conditions, amitraz (300 ppm) showed 16.6625 ± 6.8% mortality and 75.80% inhibition of fecundity, while flumethrin (80 ppm) showed 12.49 ± 4.165% mortality and 100% inhibition of fecundity against *R.* (*B.*) *annulatus* (Appendix A). Adult ticks treated with either 300 ppm of amitraz or 80 ppm of flumethrin were further used for gene expression studies.

### 3.3. De Novo Transcriptome Assembly Statistics

Following quality analysis, high-quality reads from the two libraries (untreated and amitraz treated) with an average length of 151 bp were clustered using Trinity, a short-read assembling program. The assembled contigs produced 50,591 (untreated) and 71,711 (treated) unigenes with a 662 and 542 bp average length, respectively. The entire identified transcripts were 76,491 for the untreated sample and 52,471 for the treated sample. The maximum contig lengths obtained in the untreated and treated samples were 15,075 and 15,881, respectively, and the minimum contig length obtained was 201 for both samples. The N50 values were 1169 for untreated samples and 731 for treated samples. The summary of the De novo Assembly Statistics and the Clusters of Orthologous Groups (COGs) Statistics are in Appendix A.

### 3.4. Transcriptome Annotation

To understand the putative protein functions, the COGs database was used to forecast and categorize the potential functional involvement of the identified unigenes. Since a reference genome was not available, a sequence similarity search was performed with all available arthropod protein sequences (UniProt Database) to identify homologous genes in other species putatively. Annotation of transcripts was conducted using the BLASTALL program and only transcripts with more than 30% identity were considered for analysis. A total of 92,341 transcripts were generated; of these, there were 35,585 annotated transcripts and 56,756 unannotated transcripts. BLAST searching of non-redundant databases provided important matches, allowing the gene families most likely encoding gene products engaged in xenobiotic sequestration and detoxification to be curated and annotated.

Annotation with BLAST2GO generates a functional classification of the identified *R.* (*B.*) *annulatus* unigenes based on sequence homology. The Gene Ontology database was used to classify the functions of *R.* (*B.*) *annulatus* unigenes into three main categories associated with molecular function, cellular component formation, and the biological process. Those were further categorized into nine biological processes, seven cellular components, and ten molecular functions, respectively (Figure 3). Among the nine subcategories of biological process, DNA integration occupied the highest percentage (6%) followed by the transmembrane transport (1.49%), regulation of transcription (1.16%), and intracellular signal transduction (1.16%). The major subcategories within molecular function included nucleic acid binding (19.26%), followed by ATP binding (8.41%) and zinc ion binding (8.32%). In the cellular component domain, the majority of the GO terms were specific to the integral component of the membrane (10.75%), followed by the nucleus (4.3%). Our Gene Ontology analysis of the *R.* (*B.*) *annulatus* showed that a high percentage of genes are in the molecular function category, of which about 19.26 % are involved in nucleic acid binding.

### 3.5. KEGG Pathway Analysis

The Kyoto Encyclopedia of Genes and Genomes (KEGG) database gathers manually drawn pathway maps, allowing pathway-based analysis to understand gene interactions and biological functions. Using the KASS server, the identified unigenes were functionally assigned to several different pathways specific to *R.* (*B.*) *annulatus* tick species, although a total of 38,472 pathways were identified. These unigenes were highly represented in metabolic pathways, signal transduction pathways, cellular community, and transport and catabolism pathways (Figure 4). These annotations provided insight for inquiring into a particular cellular processes, molecular functions, and metabolic pathways in the *R.* (*B.*) *annulatus*.

### 3.6. Identification of Simple Sequence Repeats

Due to the high degree of polymorphism, abundance, and ease of development [58], Simple Sequence Repeats (SSRs) have been routinely used as molecular markers for establishing genome diversity across species. To identify SSRs, the assembled results of transcripts were screened with the MISA search tool. A total of 9234 sequences were examined, which had a total sequence size of 58,009,214 bp, and 18,749 potential SSRs were identified from the *R.* (*B.*) *annulatus* transcriptome data. The identified SSRs were predominantly mononucleotide and tetranucleotide repeats, representing about 63% (11,769) and 18% (3457) of the SSRs, respectively. A small fraction of dinucleotide repeats (12%) and pentanucleotide repeats (7%) were also identified in the *R.* (*B.*) *annulatus* transcriptome. Features of SSRs identified in the *R.* (*B.*) *annulatus* transcriptome are shown in Appendix A.

### 3.7. Transcripts Encoding Detoxification Enzymes and Insecticide Targets

A salient feature of metabolic resistance to acaricides is when the highest transcriptional expression is of detoxifying enzymes, which usually results in increased activity and significant overexpression of CYP450s, Esterases (ESTs), and Glutathione S-transferases (GSTs) genes. Previously published insect genome data of *Anopheles gambiae*, *T. castaneum*, and *Drosophila melanogaster* [59,60,61] were mined to retrieve genes encoding various detoxification enzymes, which were used for a homology search in *R.* (*B.*) *annulatus*. In our study, a total of 70 detoxifying genes were identified from the *R.* (*B.*) *annulatus* transcriptome, including 44 CYP450 genes, 23 GST genes, and 3 EST genes encoding detoxifying enzymes. These detoxifying genes were differentially expressed when triggered by amitraz (Figure 5). The genes differentially upregulated after feeding were selected and discussed as potential antigen candidates for tick vaccines.

### 3.8. Comparative Transcriptomic Analysis of Acaricide Treated and Untreated R. (B.) annulatus Larvae

RNA-seq is a powerful and seminal technology used to gauge the gene expression response at the tissue, organism, and whole-genome level. Differentially expressed contigs between the amitraz-treated and untreated *R.* (*B.*) *annulatus* tick larvae were determined using the DESeq and Bioconductor (http://bioconductor.org/ (accessed on 25 August 2015) R-packages (Figure 6). To examine whether amitraz treatment against *R.* (*B.*) *annulatus* larvae resulted in statistically significant changes in gene expression, the quantum of gene expression was analyzed after normalizing the gene abundance from each library to reads per kb per million reads (RPKM). Comprehensive DESeq computational analysis facilitated the recognition of differentially expressed transcripts vis-à-vis amitraz. Overall, 16,635 transcripts exhibited upregulation, 15,539 exhibited downregulation, and 56,864 transcripts were neutral. Notably, 1898 transcripts showed expression only in untreated conditions, whereas 1403 showed expression only in treated conditions. The complete transcriptome data revealed substantial differences in the expression patterns among amitraz-treated and untreated *R.* (*B.*) *annulatus* larvae. We also compared the differentially expressed detoxifying genes upon amitraz treatment, which revealed 29 upregulated genes, 12 downregulated genes, and 27 genes that were neutral (Table 1). Compared to the untreated sample, one CYP450 gene was expressed only in the treated condition. One CYP450 gene was expressed in then-untreated condition but did not show any expression in response to amitraz treatment.

### 3.9. Gene Expression Profile of Metabolic Detoxifying Genes in Acaricides Treated R. (B.) annulatus Larvae and Adults Using RT PCR

Several metabolic detoxifying genes identified from the RNAseq data were further used for expression studies with Real-Time PCR. The genes were named using their respective family name followed by the transcriptome data ID. We selected nine detoxifying genes (Appendix A) to analyze their upregulation/downregulation in various developmental stages of *R.* (*B.*) *annulatus* and in response to different insecticides. Genes such as CYP (30980) and CYP (41222) (Figure 7A) and GST (34065) (Figure 7B) were significantly upregulated in response to amitraz treatment at both adult and larval stages, making them important candidate genes for drug resistance against amitraz. However, CYP (51361) and CYP (38633) (Figure 7C) showed significant overexpression in response to flumethrin treatment at both adult and larval stages. A few genes were only upregulated in the ticks at adult stages in response to flumethrin, such as CYP (41222) and CYP (36267). The GST gene 55171 showed higher expression in flumethrin-treated larvae (Figure 7D). In our study, RT-PCR results were consistent with the transcriptomic data obtained from the tick larvae treated with amitraz.

## 4. Discussion

*Rhipicephalus* (*Boophilus*) *annulatus* ticks are adept at transmitting a large variety of infectious diseases, such as anaplasmosis and babesiosis, and at causing huge impediments in the cattle industry. *R.* (*B.*) *annulatus* reside in large areas of Southern India [10,45]. *R.* (*B.*) *annulatus* voraciously sucks blood meal, injects toxins, transmits pathogens, and finally modifies immune responses in cattle, which negatively impacts cattle productivity and results in significant economic losses [62,63]. Tick control through acaricide treatment remains a common strategy, although progress toward vaccine development and other alternative tick control methods has also been made. Amitraz, a formamidine pesticide predominantly used in ectoparasite control in several geographic areas of South India, is frequently used due to its inhibitory effect on tick fecundity and its impairment of oocytes [44,45]. However, the high efficacy of amitraz against ticks has diminished due to the development of acaricidal resistance. Consequently, *R.* (*B.*) *annulatus* with high resistance levels are insensitive to certain amitraz formulations, rendering this acaricide method ineffective for tick control. Thus, it is necessary to understand the mechanism through which *R.* (*B.*) *annulatus* counteracts the effect of insecticide. Ongoing research and development initiatives are geared toward finding a successful management strategy for this pest.

Previous studies on the underlying resistance mechanism in ticks led to the understanding that behavioral, biochemical, and genetic mechanisms are involved in neutralizing acaricide [34,64,65,66]. The genetic mechanisms accelerating resistance evolution may involve target site insensitivity or increased metabolic enzymes, such as cytochrome P450s, GSTs, CCEs, and ABC transporters, which can sequester, catalyze, or aid the elimination of the acaricide molecules, thus inhibiting them from binding their target [67]. The extensive scientific background on metabolic enzymes [43,68,69] encouraged us to assess the expression levels of metabolic detoxifying enzymes in response to amitraz. In this study, a transcriptomics approach was employed to obtain the differential expression levels of genes related to metabolic detoxification. In addition, qRT-PCR was used to examine the expression levels of CYPs and GSTs in *R.* (*B.*) *annulatus* at sublethal doses of flumethrin and amitraz.

In vitro bioassays, such as the larval packet test and adult immersion test, were conducted to compare the acaricidal effects of amitraz and flumethrin against *R.* (*B.*) *annulatus*. Larval toxicity experiments revealed that *R.* (*B.*) *annulatus* showed a slight susceptibility difference against amitraz and flumethrin, indicating that flumethrin is comparatively more effective at larval mortality than amitraz. The LC_50_ value observed was 100 ppm for amitraz and 80 ppm for flumethrin against *R.* (*B.*) *annulatus* adult ticks. Based on this result, flumethrin seems to be comparatively effective to amitraz. This result is similar to the acaricidal efficacy of flumethrin in *R.* (*B.*) *microplus* collected from Northern India, which showed high efficacy compared to other compounds [70]. The results presented here, along with previous data, advocate that a lower concentration of flumethrin is required to produce 50% mortality compared to amitraz. This may be due to the late introduction of flumethrin into the Indian market [70]. Several previous studies on the efficacy of amitraz and flumethrin have revealed a different spectrum of susceptibility in tick populations [45,71,72]. This susceptibility may be influenced by many factors such as the application method, dose, frequency of acaricides application, genetic variation of the tick, geographic locations, defense mechanisms in the cattle, and the breed of the cattle [73,74]. Considering these factors, livestock farmers can logically decide to adopt a particular chemical for tick control. Although several studies have assessed the acaricidal efficacy of different compounds against *R.* (*B.*) *annulatus* [43,45,70], our understanding of the acaricidal resistance mechanism at the genetic level is limited due to the lack of genomic information of this ectoparasite. Therefore, to better understand the acaricidal resistance molecular mechanism and to associate metabolic enzymes with acaricide resistance, we conducted a comparative transcriptome analysis of amitraz treated and untreated *R.* (*B.*) *annulatus*. We used the larval stage of *R.* (*B.*) *annulatus* for de novo sequencing, assembly, annotation, and downstream analysis. According to the annotated results, 70 genes were involved in the detoxification mechanism in *R.* (*B.*) *annulatus*. Among these 70 genes, 44 were annotated as CYP450, 23 as GSTs, and 3 as ESTs. Considering the abundance of detoxifying genes, it was intriguing that more than half of the genes belonged to the Cytochrome P450 monooxygenase family. To further investigate gene expression changes associated with the amitraz treatment in *R.* (*B.*) *annulatus,* differentially expressed contigs were compared using DESeq. A total of 16,635 transcripts were upregulated and 15,539 were downregulated. Among the results obtained from the DESeq analysis, we compared differentially expressed detoxifying genes upon amitraz treatment. Of these, 29 upregulated (11 CYP450 and 18 GST) and 12 downregulated (11 CYP450 and 1 GST) detoxifying genes were identified. Further, a total of nine differentially expressed genes obtained from the Illumina analysis were selected to examine expression changes during various developmental stages of *R.* (*B.*) *annulatus* mediated by amitraz and flumethrin using qRT-PCR. Among the 9 genes, 3 CYP genes CYP (41222), CYP (27576), and CYP (30980) and 1 GST gene, GST (34065) were highly expressed in both amitraz-treated *R.* (*B.*) *annulatus* adults and larvae. Similarly, for the flumethrin-treated samples, 3 CYP genes CYP (38633), CYP (51361) and CYP (30980) were significantly upregulated in adults and larvae.

Cytochrome P450 Monooxygenases are well known for oxidizing widely diverse substrates and are capable of producing an array of molecular reactions [75]. The existence of the cytochrome P450 system has been established in different arthropod species, including ticks. Previous studies on metabolic resistance showed that the highest expression of cytochrome P450-like transcripts promotes coumaphos-resistant Mexican populations of *R.* (*B.*)*microplus* [76]. Different expression patterns, as well as expressional variations of transcripts encoding CYP450 genes, in multiple tissues and during different developmental stages indicate CYP450 catalytic activity [77]. As a result, populations of cattle ticks, *R.* (*B.*)*microplus*, have developed tolerance to nearly all synthetic and chemical acaricides over the past few decades due to their intensive use [78]. Comprehensive analysis of the expression of P450 cytochrome oxidases in *R.* (*B.*) *microplus* -associated pyrethroid acaricide resistance with the level of CYP expression [34]. Studies [43,68,69,79,80] have also reported the association of acaricide resistance in ticks with an increased level of expression of P450s genes, specifically due to their role in xenobiotic detoxification. Apart from CYP protein, high expression of GSTs has also been identified, which plays a role in conferring resistance to ticks [65,81]. In our study, the high expression of CYP and GST genes in *R.* (*B.*) *annulatus* during the acaricide-stressed condition might implicate their role in resistance development and the increased survival rate of *R.* (*B.*) *annulatus*. These genes could be used as targets for pest control, such as by inhibiting their action through sequence-specific gene silencing via RNA interference [82]. Additionally, the downregulation of certain genes in response to acaricides could be due to other novel feedback mechanisms. Understanding these pathways could open new avenues to improve tick management strategies.

The larval insensitivity to flumethrin may indirectly highlight its efficiency as a drug used to target the larval stage compared to amitraz. This is consistent with the larval packet and adult immersion tests, where flumethrin caused more lethality than amitraz. This could be attributed to the unaltered gene expression pattern of various metabolic resistance genes in the larval stages of the ticks. This provides evidence that overexpression could confer acaricide resistance, possibly in the absence of target site insensitivity.

Overall, our research exploited a combination of transcriptome analysis, differential gene expression, and gene expression validations to study the acaricide resistance mechanism. These results showed that our approach relying on the Illumina Hi-Seq platform enabled a comprehensive representation of the *R.* (*B.*) *annulatus* transcriptome. The identified gene targets from this study may be used as valid targets for the currently evolving gene-based tick management strategies. These outcomes are geared to evaluate the possible molecular mechanisms involved in tick-borne diseases, thus positively benefiting medical and veterinary professionals, farmers, and the common public. The knowledge gained from this study has the potential to aid practical applications, especially when generating new strategies to control menacing ticks.

## 5. Conclusions

An improved understanding of the detoxifying genes in *R.* (*B.*) *annulatus* is indispensable to perform a functional analysis to understand the complex processes involved in acaricide resistance, thus providing valuable insights into the various pathways and processes that are crucial for the survival of ticks. This study quantified the relative expression of detoxifying genes in insecticide-treated and untreated *R.* (*B.*) *annulatus* and correlated their expression using qRT-PCR and Illumina sequencing to discern the genetic and molecular basis of insecticide resistance. Having realized the overriding significance of these genes in *R.* (*B.*) *annulatus*, we have analyzed their expressional variations at two stages: the larval and adult stages, in response to two different acaricides. Increasing access to Next-generation Sequencing technology, as well as transcriptome technology, can accelerate research on acaricide resistance and could play a significant role in developing novel tools to control the tick population. Taken together, this will propel further research in the field of toxicology and insecticide resistance, as well as advance our understanding of the role of environmental adaptation and microevolution as vital processes underlying resistance development.

## Figures and Tables

**Figure 1 biomedicines-11-01369-f001:**
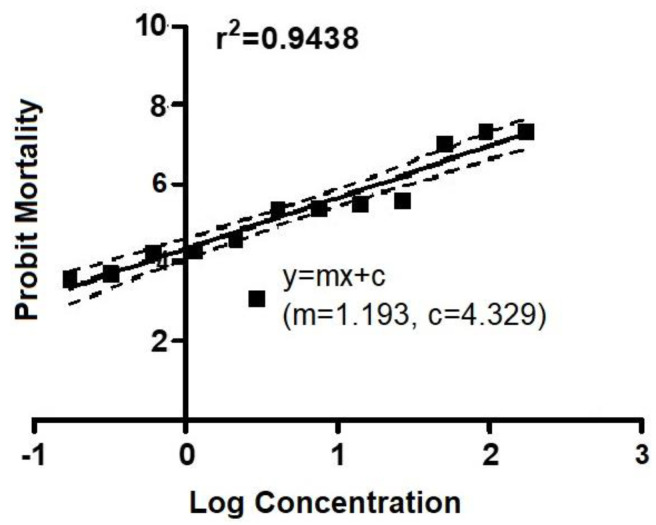
Dose–response mortality data *R.* (*B.*) *annulatus* larvae against amitraz using LPT.

**Figure 2 biomedicines-11-01369-f002:**
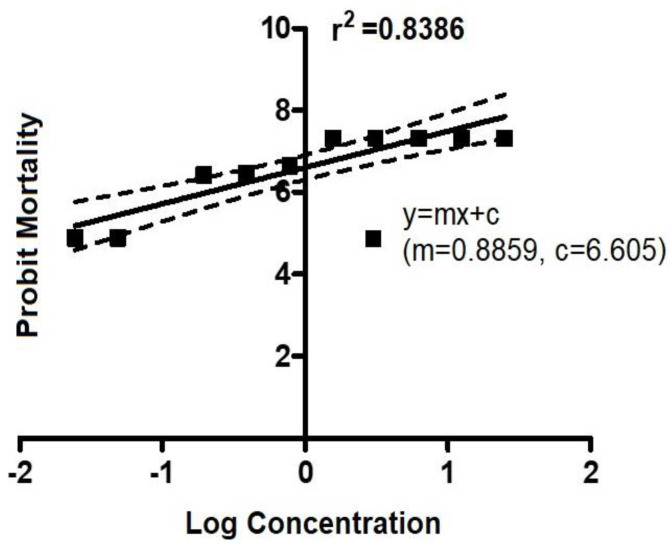
Dose–response mortality data *R.* (*B.*) *annulatus* larvae against flumethrin using LPT.

**Figure 3 biomedicines-11-01369-f003:**
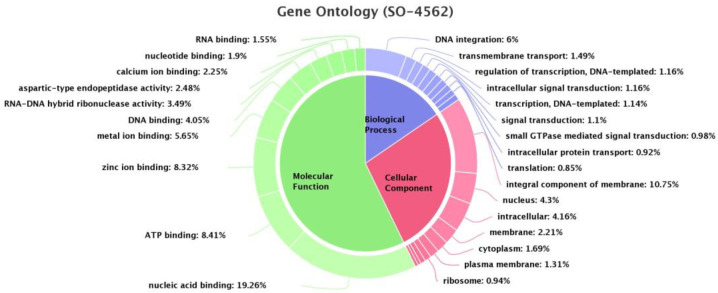
Summary of the Gene ontology classification from the *R.* (*B.*) *annulatus* transcriptome.

**Figure 4 biomedicines-11-01369-f004:**
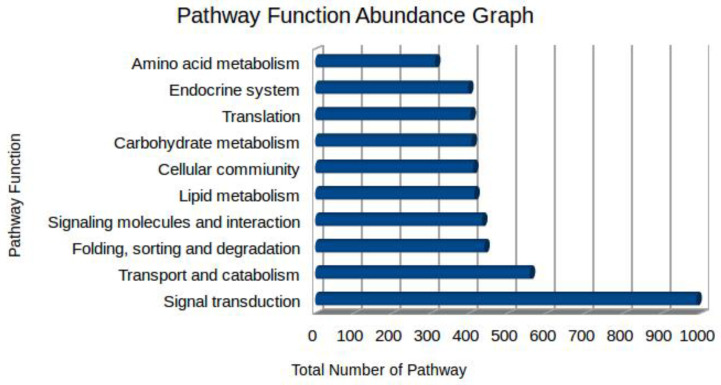
Pathway function abundance graph from the *R.* (*B.*) *annulatus* transcriptome.

**Figure 5 biomedicines-11-01369-f005:**
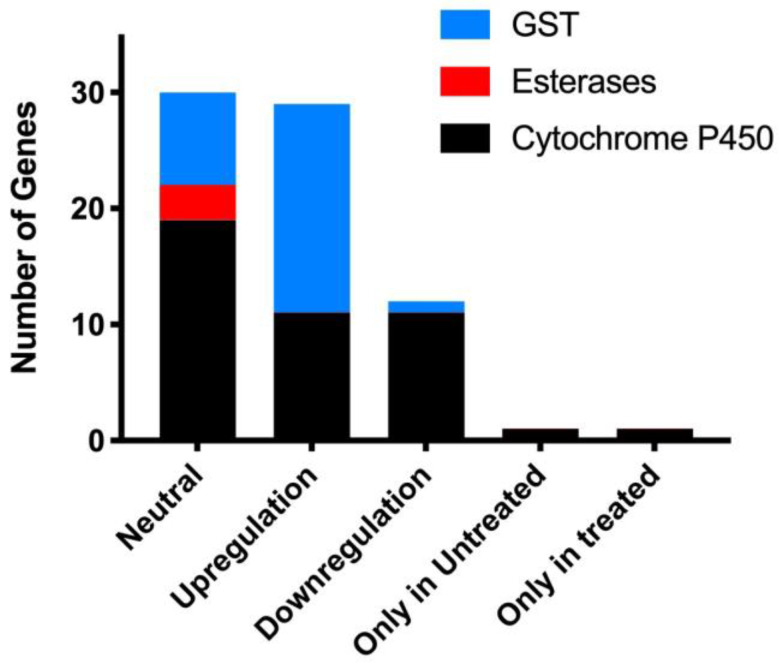
Number of differentially expressed detoxifying genes identified from the *R.* (*B.*) *annulatus transcriptome*.

**Figure 6 biomedicines-11-01369-f006:**
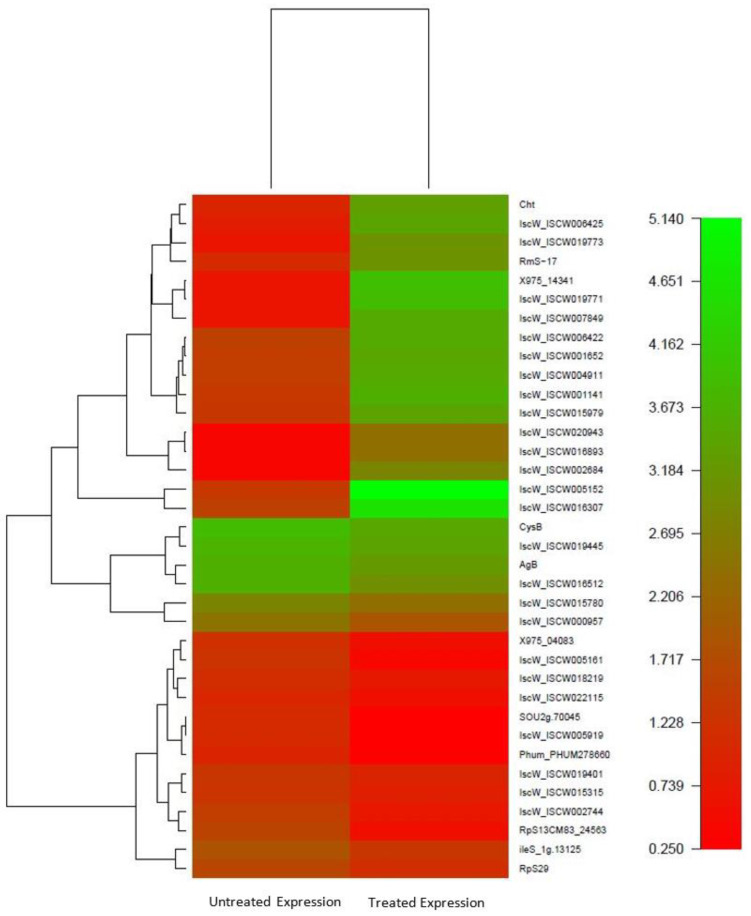
Heatmap of differentially expressed genes in untreated and amitraz treated *R.* (*B.*) *annulatus* larvae.

**Figure 7 biomedicines-11-01369-f007:**
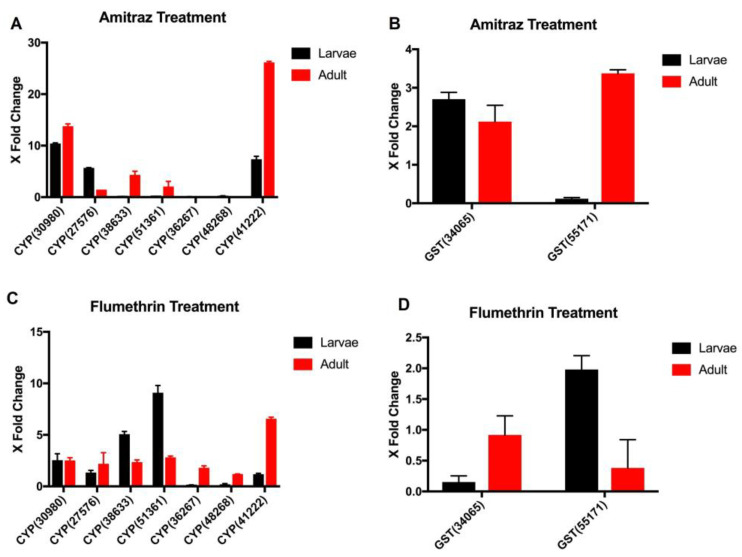
Stage-dependent expression patterns of metabolic detoxifying genes in different acaricide treated *R.* (*B.*) *annulatus* (**A**) Expression patterns of CYP450 genes in response to amitraz. (**B**) Expression patterns of GST genes in response to amitraz. (**C**) Expression patterns of CYP450 genes in response to flumethrin. (**D**) Expression patterns of GST genes in response to flumethrin. The data are presented as foldchange in expression level in drug-treated ticks relative to the gene expression level in untreated ticks.

**Table 1 biomedicines-11-01369-t001:** Differential gene expression of detoxifying genes in *R.* (*B.*) *annulatus*.

Gene Names	No. of Upregulated Genes	No. of Downregulated Genes	No. of Neutrally Regulated Genes	No. of Genes Expressed Only in Untreated Condition	No. of Genes Expressed Only in Treated Condition
*Cytochrome P450*	11	11	16	1	1
*Glutathione–S-Transferases*	18	1	8	0	0
*Esterases*	0	0	3	0	0

## Data Availability

Data are contained within the article and Appendix A.

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
