# Peer review of "Transcriptome Profiling of Rhipicephalus annulatus Reveals Differential Gene Expression of Metabolic Detoxifying Enzymes in Response to Acaricide Treatment"

_biomedicines, 2023, doi:10.3390/biomedicines11051369_

Round 1

Reviewer 1 Report

Dear authors,

Your manuscript “Transcriptome profiling of Rhipicephalus annulatus reveals differential gene expression of metabolic detoxifying enzymes in response to acaricide treatment” is very interesting and devoted to transcriptome profiling of Rhipicephalus annulatus.

But in current form, the manuscript can’t be published.

First and main - English should be improved. The text is written very heavily and is very difficult to read and understand.

Abstract should be rewritten (repeats like “In this study, high quality RNA-seq data 17

of untreated and amitraz treated R. (B.) annulatus were assembled… In this study, high quality RNA-seq data of untreated and amitraz treated R. (B.) annulatus were assembled…” look very stgrange).

Lane 11 – Ticks are not carriers but vectors. Please, correct.

Lane 36 – “viral borne encephalitis” - absolutely wrong. There is no such term, viruses couldn’t be the vectors of diseases.

Lane 66 – “ixodide” – please, correct.

Lane 82 – “In many cases, Ticks…” – Why are you capitalizing here?

Lane 92 – You use “Amitraz” or “amitraz” throughout the text, what is correct name?

Lane 108 – What do you mean under “molecular information”?

Section 2.2 - It is unclear under what conditions the ticks were cultivated. Did they need food to go from larva to adult? If so, you need to add all conditions here.

Lane 160 – “R. (B) annulatus” – should be in italic.

Lane 163 – “FASTQC” – should be stated, what is it – program pack? Script? Something else?

Section 2.8 - It is not clear that you used 1 pair of primers for each gene for performing quantitative PCR (as I can tell from the supplement)? If so, how did you do the detection? Was it a probe, or something else?

Reviewer 2 Report

The Manuscript [biomedicines-2175411] entitled [Transcriptome profiling of Rhipicephalus annulatus reveals differential gene expression of metabolic detoxifying enzymes in response to acaricide treatment] assembled, annotated and analyzed the high quality RNA-seq of untreated and amitraz treated R. annulatus. Authors also analyzed the differential gene expression to map the detoxifying genes. Also, examination of expression level of detoxifying genes was performed in different development stages using real time PCR.

Major comments:

1-      Line 121: [2.2. Larval Packet Test], control is missing. How authors did the control?. How authors applied these concentrations and control?. How many of concentrations for each acaricide?. Mention them exactly.

2-      Line 135: [2.3. Adult Immersion Test], the same previous comments.

3-      Was the absolute methanol used in the 2.2 and 2.3 experiments?. It is already toxic for ticks.

4-      In gene expression, authors used two different doses [LC50] on larvae and adults. Then, there are 2 factors, thus the comparison is incorrect.

5-      There is no control for studying the gene expression to compare between treated and untreated ticks.

Minor comments:

1-      Line 21-22: [In this study, high…. were assembled] it is repeat of lines 17-18, delete it

2-      In all the MS, use [R. annulatus] instead of [R. (B.) annulatus]

3-       Line 49: Add the full names of B. bovis and B. bigemina

4-      Line 131: What is this incubator and company?. Use (%) instead of [per cent]

5-      Lines 132-133: write as [The median lethal concentrations (LC50)].

6-      In Results, there are many repeats for methods.

Round 2

Reviewer 1 Report

Dear authors,

Thank you for the corrections made to the manuscript - it has become much better.

I have only a few remarks:

Text formatting - different line spacing in text, correct, please.

Lane 38 – “encephalitis” – encephalitis is the name of a disease of a different origin; in your case, you should use “tick-borne encephalitis”;
Lane 167 – “program” – “software” is better.

Reviewer 2 Report

The revised version of the Manuscript [biomedicines-2175411] entitled [Transcriptome profiling of Rhipicephalus annulatus reveals differential gene expression of metabolic detoxifying enzymes in response to acaricide treatment] did not reached all major comments where:

1-      The absolute methanol used in the 2.2 and 2.3 experiments is toxic for ticks. In these experiment it could use Methanol for extraction BUT using other safe solvent as DEMSO (1%).

2-      In gene expression, authors used two different doses [LC50] on larvae and adults. Then, there are 2 factors, thus the comparison is incorrect. It means that: HOW to compare between two findings affected by 2 factors?. THIS statistical analysis should be done by TWO-WAY ANOVA.

3-      Authors mentioned that [For all the gene expression analyses including the RNA seq and real time RT PCR, untreated ticks were used as control.] BUT, there is no any data about the control in the gene expression to compare between treated and untreated ticks.
